# Could Artificial Intelligence Prevent Intraoperative Anaphylaxis? Reference Review and Proof of Concept

**DOI:** 10.3390/medicina58111530

**Published:** 2022-10-26

**Authors:** Mihai Dumitru, Ovidiu Nicolae Berghi, Iulian-Alexandru Taciuc, Daniela Vrinceanu, Felicia Manole, Adrian Costache

**Affiliations:** 1Department of ENT, ‘Carol Davila’ University of Medicine and Pharmacy, 050751 Bucharest, Romania; 2Victoria Ambulatory Department, ‘Sfânta Maria’ Clinics and Laboratories, 011013 Bucharest, Romania; 3Department of Pathology, ‘Carol Davila’ University of Medicine and Pharmacy, 020021 Bucharest, Romania; 4Department of ENT, Faculty of Medicine, University of Oradea, 410073 Oradea, Romania

**Keywords:** allergy, anaphylaxis, drugs, intraoperative, artificial intelligence

## Abstract

Drugs and various medical substances have been used for many decades to diagnose or treat diseases. Procedures like surgery and anesthesia (either local or general) use different pharmacological products during these events. In most of the cases, the procedure is safe and the physician performs the technique without incidents. Although they are safe for use, these substances (including drugs) may have adverse effects, varying from mild ones to life-threatening reactions in a minority of patients. Artificial intelligence may be a useful tool in approximating the risk of anaphylaxis before undertaking a medical procedure. This material presents these undesirable responses produced by medical products from a multidisciplinary point of view. Moreover, we present a proof of concept for using artificial intelligence as a possible guardship against intraoperative anaphylaxis.

## 1. Introduction—Surgery and Anesthesia—Perioperative Events

Perioperative events, including anaphylaxis, may appear in both adults (more frequent) and children, typically after anesthetic induction. These reactions are classified into six types (ABCDEF): augmented (dose-related), bizarre (non-dose-related), chronic (dose-related and time-related), delayed (time-related), end of use (withdrawal) and failure (failure of therapy) [1]. The reactions may be immune (usually IgE-mediated and sometimes IgG-mediated) or non-immune in nature [2]. Adverse drug reactions (ADR) are defined as any unintended and undesired effect of a drug that occurs at doses used for prevention, diagnosis or treatment. ADR can also be broadly classified into two types: predictable and unpredictable. Predictable reactions are dose-dependent, related to the known pharmacologic actions of the drug and appear either in people with genetic variability or in normal individuals. They account for about 80% of all ADRs and are subdivided into side effects, secondary effects, overdose and drug interactions. Unpredictable reactions are not related to the dose or pharmacologic actions of the drug and occur mainly in susceptible individuals. They are divided into drug intolerance (undesirable pharmacologic effect that occurs at low or subtherapeutic doses), drug allergy and nonallergic reactions which result from direct release of mediators from mast cells and basophils [3,4].

## 2. Neuromuscular Blocking Agents (NMBA)

NMBA block neuromuscular transmission at the neuromuscular junction, causing paralysis of the affected skeletal muscles. Their action is exerted on the post-synaptic acetylcholine (Nm) receptors. These drugs fall into two groups:

※Non-depolarizing blocking agents: These agents make up the majority of the clinically relevant neuromuscular blockers. They work by competitively blocking the binding of ACh to receptors, and also by blocking directly the ionotropic activity of the ACh receptors. Non-depolarizing neuromuscular blockers can be divided into two classes based on their chemical structure: steroidal (e.g., rocuronium, vecuronium, pancuronium) or benzylisoquinoline (e.g., mivacurium, atracurium, cisatracurium);

※ Depolarizing blocking agents: These agents work by depolarizing the sarcolemma of the skeletal muscle fiber. This persistent depolarization makes the muscle fiber resistant to further stimulation by ACh [5,6].

The mechanism of the reaction may involve one or more from the following: allergic (IgE-mediated), pseudo-allergic or direct release of histamine from mast cells. The substituted ammonium ions are the sites implicated in the IgE recognition. Cross-reactivity is more frequent with NMBA amino steroids. In some situations, the reaction to NMBAs (including anaphylaxis) may appear at first exposure. This is probably due to previous exposure to compounds containing tertiary and/or quaternary ammonium groups, such as cosmetics, disinfectants, personal-care products [3].

※Steroidal NMBAs presented 15 cases of perioperative anaphylaxis, as an incidence rate of four in 10,000 was recorded in Singapore from 2015 to 2019 in a tertiary pediatric hospital from 35,361 cases of pediatric anesthesia. Rocuronium and atracarium were responsible for one case each [7]. Rocuronium was the second cause of drug-induced anaphylaxis during general anesthesia in 14 tertiary hospitals in Japan, in 46 cases of anaphylaxis at Gunma University Hospital and 13 neighboring hospitals in a retrospective, multicenter, observational study accomplished between 2012 and 2018 [8]. Rocuronium and Suxamethonium (another NMAB) were significantly more involved in perioperative anaphylaxis than the other available NMBAs in a French pharmacovigilance survey conducted from 2000 to 2012 [9]. Rocuronium was perceived as the most frequent cause of anaphylaxis from among all NMBAs in an electronic survey that included the answers of 11 104 anesthetists (77% crude response rate) from 341 (96%) hospitals in the United Kingdom [10]. The rate of anaphylaxis to succinylcholine and rocuronium was approximately 10-fold higher than to atracurium in a retrospective, observational cohort study of the rate of intraoperative anaphylaxis to NMBDs at two hospitals from Auckland, New Zealand, between 2006 and 2012 [11]. Rocuronium was responsible for 56% of cases of NMBD anaphylaxis, versus 11% caused by vecuroniummin, in the only specialized diagnostic center in Western Australia over a 10-year period. Rocuronium had a higher rate of IgE-mediated anaphylaxis compared with vecuronium (8.0 vs. 2.8 per 100,000 exposures; *p* = 0.0013) [12]. Rocuronium and suxamethonium were the most common agents for triggering anaphylaxis, according to a ten-year audit of allergy testing (2000–2009) at the Royal Adelaide Hospital from Australia [13]. Succinylcholine and rocuronium were the most frequently incriminated in an old French study that included 467 patients between 1 January 1997 and 31 December 1998 [14]. Rocuronium was associated with an increased risk of anaphylaxis, compared with other NMBAs in obese patients presenting for bariatric surgery in Western Australia between 2012 and 2020 [15]. Rocuronium was the most frequently causative NMBA of perioperative allergic reactions in a retrospective analysis of patients referred to a University Allergy Centre in Belgium, with the suspected cause being an allergic reaction during or shortly after general anesthesia [16]. Rocuronium and vecuronium presented the highest incidence of intraoperative anaphylaxis in a retrospective observational study conducted in two tertiary hospitals in South Korea between 1 January 2005 and 31 May 2014. From 729,429 patients who were exposed to NMBA, nineteen episodes were deemed to be caused by intraoperative anaphylaxis reaction to NMBAs [17]. Vecuronium was the NMBA causing the largest number of reactions in a 12-year survey at a French pediatric center in the period between 1989 and 2001 [18]. Rocuronium and vecuronium were the second and third cause of anaphylaxis a 6-year single-center follow-up study (1996–2001) that examined 83 anaphylactic reactions related to general anesthesia performed at one allergy center in Bergen, Norway [19]. Allergic/other immune side effects of pancuronium were much less described in the medical literature; in addition, two studies, one in Sydney, Australia, that used intradermal skin tests and basophil activation testing [20] and one in Cluj-Napoca, Romania that used skin tests [21] did not find a significant sensitization between pancuronium and other NMBAs.

※Benzylisoquinolinium NMBAs: Anaphylaxis to mivacurium was first described in a case report form 1996 in Auckland, New Zealand [22]. Mivacurium activated MRGPRX2 and triggered mast cell degranulation, leading to anaphylactoid reactions in an experiment using wild-type mice [23]. Atracurium and cis-atracurium (the 1R cis-1′R cis isomer of atracurium) were also involved in anaphylactic reactions. Cis-atracurium also activated MRGPRX2 and triggered mast cell degranulation, leading to anaphylactoid reactions in wild-type mice [24]. Allergy to either drug was associated with allergy to the stereoisomer, with skin testing for allergies to neuromuscular blocking drugs being only requiring the use either atracurium or cis-atracurium (cis-atracurium being the preferred drug) [25]. Atracurium was the second leading cause of anaphylaxis after rocuronium among NMBAs in the 6th National Audit Project (NAP6) on perioperative anaphylaxis collected from all NHS hospitals in the UK over 1 year. [26]. Atracurium and cis-atracurium were also implicated in the French study mentioned before [9]. A low frequency of reactions involving cis-atracurium was confirmed in another French study from 2007 [27]. Recent case reports have involved atracurium as a cause of cardiac arrest [28] and cis-atracurium besylate as a cause of serious bronchospasm [29].

※Succinylcholine (suxamethonium): Succinylcholine can cause nonimmunologic histamine release, but there have also been reports of IgE-mediated reactions in some patients [30]. Succinylcholine-induced anaphylaxis was twice as likely than the same reaction being caused by other neuromuscular blocking agents, as was determined in the 6th National Audit Project (NAP6) [26]. In a case series of life-threatening succinylcholine-induced anaphylaxis consisting of 21 case reports of life-threatening reactions, 67% of cases occurring during scheduled surgery and 33% during emergency surgery [31]. The rate of anaphylaxis to succinylcholine was the highest in the study of Reddy et al. [11]. Succinylcholine was responsible for 21% cases of NMBD anaphylaxis in the article of Sadleir et al. [12]. Dong et al. established that succinylcholine was the most frequently incriminated NMBA for anaphylaxis in their study [27]. Many unique cases have been reported for succinylcholine, such as those described by Sleth [32] and Gibbs [33].

※Sugammadex is a modified γ-cyclodextrin, with a lipophilic core and a hydrophilic periphery used as a medication to reverse the neuromuscular blockade induced by rocuronium and vecuronium [34]. The incidence of anaphylaxis to sugammendex presents important differences between authors and countries. It was the most common causative agent in a Japanese study [8] and anaphylaxis occurred in only 2 of the 19,821 patients, who received a total of 23,446 doses in a single institution cohort study [35]. The incidence of anaphylaxis caused by sugammadex was calculated as 0.02% (of the number of sugammadex cases) (95% CI: 0.007–0.044%) from a total of 49,532 patients who received general anesthesia included in the study [36]. The estimated reporting rate of anaphylaxis due to sugammadex was 0.0055–0.098% using an in-app survey that analyzed data from 2770 participants in 119 countries who responded between March 2016 and May 2017. It is estimated that between 1.6–2.9 million doses were administered [37].

## 3. Antibiotics

Perioperative anaphylaxis to antibiotics typically occurs in the first minutes after drug exposure. The most common antibiotics implicated in this reaction are β-lactams and vancomycin. There is a high cross-reactivity between the penicillin and the first-generation cephalosporins [2]. Antibiotics were the commonest culprits for anaphylaxis in Toh’s study [7]. Banerji et al. performed a retrospective review of patients, referring to the outpatient Allergy/Immunology (A/I) clinic at Massachusetts General Hospital for 57 perioperative allergic reactions between October 2003 and May 2017. One causative agent was identified in 28 patients out of 123 patients referred. Seventeen of 28 (61%) patients were positive to an antibiotic, including 13 (46%) positives to cefazolin, 2 to ciprofloxacin and 1 to vancomycin and levofloxacin [38]. Wakimoto and colleagues reviewed the Wake-up Safe database from 2010 to 2017 and identified all reported instances of anaphylaxis. Among 2,261,749 cases reported to the Wake-up Safe database during the study period, perioperative anaphylactic reactions occurred at a rate of 1: 36,479 (0.003%). There has been 14 described cases of antibiotic allergy [39]. A retrospective chart review of referrals to a tertiary allergy clinic from Hong Kong to Queen Mary Hospital drug allergy clinic in the period 2012–2016 presented 60 cases of allergic reactions. Antibiotics ranked the second place after NMBAs (14 cases). All the antibiotic allergy cases were caused by beta lactams, most commonly cefazolin, a common choice of antibiotic therapy in that region [40]. An extensive review from 2019 placed β-lactam antibiotics such as amoxicillin/clavulanic acid, cefazolin, and cefuroxime as the most significant cause of perioperative anaphylaxis from the antibiotics followed by teicoplanin and vancomycin [41]. Perioperative hypersensitivity was evaluated in a 17-year survey in Antwerp, Belgium. A total of 608 cases formed the final dataset. Overall, 67 antibiotic allergies were identified, accounting for 15.3% of allergic reactions, with cefazolin and amoxicillin/clavulanic acid being the most frequent [42]. Beta-lactam antibiotics (penicillin and cephalosporins), vancomycin and quinolones were the drugs from this category most frequently involved in adverse reactions during the perioperative period representing the third cause of anaphylactic reactions in surgical patients. Various mechanisms, both IgE-mediated and non-IgE-mediated, are involved in adverse drug reactions due to penicillin and other beta lactams such as cephalosporins in addition to unknown mechanisms. Vancomycin, a glycopeptide antibiotic used for patients known with penicillin allergies and in the cases of Gram-positive resistant organisms, may provoke reactions both the IgE-mediated and non-IgE-mediated. Vancomycin is the agent responsible for “red man syndrome” representing a non-IgE-mediated hypersensitivity reaction characterized by flushing, pruritus, an erythematous rash of the head and upper torso, and arterial hypotension. These reactions appear along with the rapid infusion of the drug at the first dose. Quinolones represent the third most important group of antibiotics involved in perioperative anaphylaxis. They can induce hypersensitivity reactions mediated by IgE, T cells and also non-IgE-mediated reactions. IgE-mediated reactions are more common and are severe in most cases [43]. Co-amoxiclav, teicoplanin and cefuroxime were the three most prevalent antibiotics in the 6th National Audit Project (NAP6). Gentamicin, flucloxacillin, piperacillin and tazobactam, vancomycin and metronidazole were also identified as triggers [26]. Teicoplanin is a first-generation glycopeptide antibiotic use in Gram-positive bacterial infections. Teicoplanin is used in patients labeled as penicillin-allergic as a second-line therapeutic treatment and a first-line prophylactic treatment within perioperative settings in the United Kingdom. A total of 418 unique individuals were evaluated for perioperative anaphylaxis and hypersensitivity reactions at the Department of Adult Allergy at Guy’s and St Thomas’ NHS Foundation Trust between 2013 and 2018. Forty-one (6%) of the teicoplanin-exposed patients completed diagnostic testing for acute-onset teicoplanin hypersensitivity. Acute-onset teicoplanin hypersensitivity was considered suspected (after positive skin prick test (SPT)/intradermal test (IDT) result) or confirmed (after positive DPT result) in 30 of 41 (73%) patients investigated [44]. Cefazolin may be responsible for immediate severe allergic reactions in the perioperative period. The particularity of this antibiotic is that patients with IgE-mediated hypersensitivity reactions to cefazolin can tolerate other beta-lactams. This fact is explained by the particular chemical structure, whose R1 side-chain are different from other beta-lactams [45]. Bacitracin was responsible for anaphylaxis and cardiovascular collapse in an ambulatory surgery center setting within 1 min after irrigation with bacitracin solution, after removal of breast implants in situ, for bilateral breast implant exchange [46]. Antibiotics were the second cause of anaphylactic reactions during anesthesia in a survey that received answers from 242 anesthesiologists form India in 2017 [47].

## 4. Hypnotics

Hypnotics can cause anaphylaxis in some situations. Barbiturates, particularly thiopental, were involved over time in cases of perioperative anaphylaxis. Direct release of histamine and specific IgE has been implicated as mechanisms of action [2,43]. Ketamine, involving both immunologic and nonimmunologic mechanisms, was responsible for a periprocedural anaphylaxis in a 2-year-old girl [48]. Etomidate was also responsible for intradermic positive tests in a study that included 14 patients with clinical records of anaphylactic reactions under general anesthesia [49]. Propofol is responsible for IgE-mediated reactions due its two isopropyl groups and direct release of histamine [2]. Propofol was involved in 1 case from 50 child patients in a retrospective study who underwent skin testing for general anesthesia between 2007 and 2019 [50]. A Mexican study ranked propofol as the third largest cause of perioperative allergic reactions in a retrospective, cross-sectional, descriptive study of patients treated at the Allergy and Clinical Immunology Department of Hospital General de Mexico [51].

## 5. Narcotics

The main type of reaction involved in the producing of perioperative anaphylaxis is the direct release of mast-cell mediators. Rarely IgE-mediated reaction is responsible for reaction in case of morphine and fentanyl [2]. Pholcodine, an opioid cough suppressant, is a risk factor for perioperative anaphylaxis in patients presenting for bariatric surgery [15]. Morphine and pholcodine are substances which at physiologic pH contain substituted ammonium groups similar to the groups present on the NMBAs. Morphine contains one amine group, while pholcodine contains two such groups, of which one is more predominantly protonated at physiologic pH [52]. Measurement of IgE to both pholcodine and morphine is useful in the evaluation of allergy to NMBAs according to a retrospective study carried out at the Royal North Sore Hospital Anesthetic Allergy Clinic (Sydney, NSW, Australia) from 2009 to 2019, covering 801 consecutive patients with 255 having skin test results for NMBAs [53].

## 6. Colloids/Plasma Expanders

Dextran, hydroxyethyl starch (HES), gelatin and albumin are widely used in surgery. All colloid fluids are associated with a risk of allergic reactions, which usually appear soon after the infusion has been initiated [54,55]. Dextran was the cause of anaphylactic shock (hypotension, tachycardia, desaturation) for a case during general anesthesia [56]. A basophile activation test with significant upregulation of CD63 and a provocation test were used to prove that HES was the cause of a severe anaphylactic reaction during anesthesia in a 15-Year-Old Boy [57]. Gelatin or gelatine (from Latin: gelatus meaning “stiff” or “frozen”) is a translucent, colorless, flavorless food ingredient, commonly derived from collagen taken from animal body parts [58,59]. Gelatin was responsible for anaphylaxis after its injection into vertebral bone under pressure in two children with spinal deformity undergoing posterior spinal fusion procedures [60]. A past tick bite and subsequent development of an allergic reaction to mammalian protein products, most notably red meat, may pose a high risk to the patient if gelatin is used during surgery [61,62]. IgE-mediated anaphylaxis to albumin provoked intraoperative hypotension during exposure for planned T5-L4 posterior spinal fusion shortly after infusion in an adolescent patient [63]. As a rule, one may expect one allergic reaction in 500 infusions for colloids [54].

## 7. Aprotinin

Aprotinin is an inhibitor of serine proteases used to diminish the risk of perioperative blood loss and the necessity for blood transfusion during coronary artery bypass graft surgery [64]. Aprotinin was the cause of a vasoplegic syndrome refractory to catecholamines and vasopressin, associated with intraoperative anaphylaxis during cardiac surgery [65]. This bovine peptide antifibrinolytic agent was proved to be responsible for a case of fatal anaphylaxis, demonstrated by the presence of specific serum IgE and serum tryptase measurements. Re-exposure to aprotinin has been estimated to have a risk of 2.8% of an allergic reaction if reused within a 6-month period [66].

## 8. Other Agents

### 8.1. Non-Steroidal Anti-Inflammatory Drugs (NSAIDs)

Metamisole was responsible for a number of cases of severe anaphylactic reactions in the early 2000s in Europe and South America. Both important mechanisms of allergic reaction were implicated: non IgE, possible IgE [2,67,68].

### 8.2. Local Anesthetics

Local anesthetics stabilize neuron membranes by inhibiting the movement of ions required to transmit neural impulses. They are available in multiple forms including gels, ointments, sprays, solutions and injectable forms. It is believed that true allergy to local anesthetics accounts for less than 1% of all reactions to local anesthetics [2,69].

### 8.3. Dyes

Fluorescence in medicine is used as a non-destructive means to track or analyze biological molecules through fluorescent emission at a specific frequency. The triarylmethane dyes, patent blue V and isosulfan blue, share a similar structure and are frequently used for lymphatic mapping. Exposure to blue dye in cosmetics and other everyday products it is thought to contribute to sensitization. The mechanism of reaction to the dye is not clear, with both specific IgE sensitization and direct mast cell activation having been described. Reactions to dyes are often delayed, occurring 30 min after injection, and can be prolonged [69,70]. A case of allergic reaction to patent blue in a patient who underwent left mastectomy with sentinel lymph node was noticed 25 min after the dye injection with increased heart frequency and skin reaction [71]. Indigocarmine (5,5 sodium salt of indigodisulphonic acid or indigotine) is a blue-colored dye which is widely used to identify the ureteral orifice in urologic procedures. Hemodynamic effects including hypertension, bradycardia and atrioventricular block have been reported in intravenous application of indigocarmine in 2 cases [72], with the additional including of hypotension and anaphylaxis, in one patient. Administration of intravenous sodium fluorescein for the assessment of ureteral efflux provoked a case of anaphylactic shock in a patient with a history of frequent severe allergic reactions [73]. The use of vital blue dyes such as isosulfan blue carries the risks of skin manifestations, interference in pulse oximetry saturation and anaphylaxis in a small number of cases, according to Haque and Nossaman [74].

### 8.4. Chlorhexidine

Chlorhexidine is a standard skin antiseptic and disinfectant. It is widely used in medical, procedural, and surgical settings. Sensitization to chlorhexidine may occur from over-the-counter disinfectant solutions and home products, such as mouthwash, toothpaste, dressings, ointments. Systemic allergic reactions have been reported to chlorhexidine via topical skin application, ophthalmic wash solution, chlorhexidine bath, coated central venous catheter, and urethral gels [69]. A number of case reports and small series of cases support this assertion. A severe, blistering reaction associated with significant pain has been associated with topical application of chlorhexidine to a 27-year-old woman [75]. Five male patients experienced perioperative anaphylactic reactions during renal transplantation because of intraoperative insertion of chlorhexidine-coated central venous catheters. Skin testing was positive in four of five patients and in vitro test results for chlorhexidine-specific IgE were positive in all the patients [76]. A 53-year-old man who was the subject of a right knee arthroscopy experienced first a pruritic rash that appeared on the chest, abdomen and limbs (12 min after administration); after 15 min, bilateral angioedema of the eyelids and feelings of shortness of breath, caused by chlorhexidine, began to be experienced [77]. Anaphylaxis to chlorhexidine-impregnated CVCs (central venous catheter) was described in four patients undergoing anesthesia for cardiothoracic procedures. Anaphylaxis was preceded, in all of the cases, by insertion of a central venous catheter (CVC) impregnated with silver sulphadiazine and chlorhexidine. Allergen-specific IgE testing (ImmunoCAP^®^) to chlorhexidine was also positive in all cases [78].

### 8.5. Radiocontrast Media (RCM)

Possible mechanisms for reaction to RCM are IgE-mediated, non-IgE-mediated, and may occur both immediately and after some delay. There are some factors that might put some patients at a higher risk: asthma, chronic kidney disease, β-blocker user, therapy with cytokine IL-2, cardiac disease, history of drug or food allergy [2]. Mertes et al. conducted extensive research of anaphylactoid reactions during anesthesia from 1981 to 2010 in English and French medical journals. They found 11 cases [79].

### 8.6. Latex

Latex comes from a sap found in rubber trees, *Hevea brasiliensis*, which is used to produce many objects we use in the modern world. A latex allergy is one of the most frequent causes of anaphylaxis in surgery and has increased in prevalence concomitant with the increased use of latex gloves to prevent transmittable infections starting in the 1980s and reached its peak during the 1990s, mostly due to the implementation of high hygiene standards after the human immunodeficiency virus (HIV) epidemic [41,80]. Several populations at risk have been identified including children with spina bifida, those with a history of multiple surgeries, especially during childhood, along with healthcare workers and non-healthcare workers frequently exposed to natural rubber latex [41]. Recently, a number of case reports and series of cases illustrated the current situation in terms of latex allergy. A 12-year-old girl, with multiple surgical histories, developed anaphylactic shock during the pyeloplasty for ureteropelvic junction restenosis [81]. A latex allergy may be an important issue during obstetric and gynecologic procedures due to the prolonged exposure of genital mucosa. Co-administration of oxytocin and vasopressin might stimulate latex-induced anaphylaxis [82]. A case of pediatric anaphylaxis (severe hypotension, tachycardia and bronchospasm) was described in a healthy 4.5-year-old Caucasian boy during surgery for congenital sursoadductorius strabismus. The particularity of the case was that the patient had no history of latex allergy, only an indirect exposure due to his mother’s occupation (cosmetician working from home) [83]. A 46-year-old man, working in dentistry, with no previous medical history including allergic reaction presented a perioperative anaphylactic shock during living donor liver transplantation that necessitated inclusive extracorporeal resuscitation [84]. A case of anaphylactic shock and myocardial infarction with ST-elevation at the same time occurred several minutes after the introduction of latex into the surgical field in a 43-year-old patient during a planned inguinal hernia surgery. The authors made a diagnosis of Kounis syndrome, a well-recognized clinical scenario that describes the coincidental occurrence of allergic or hypersensitivity reactions with acute coronary syndrome. The medical history revealed that in the past the patient had noticed an itchy rash on his hands after wearing latex gloves [85].

## 9. Artificial Intelligence Guardship

Artificial intelligence (AI) has recently become an increasingly useful tool, as humanity has begun to focus more on it. It is important that this technology is used ethically and with a high degree of responsibility [86]. To be able to build such software, some simple parameters will be needed. The parameters will be statistically correlated with the subject of interest [87].

Due to the fact that anaphylactic reactions occur within minutes of exposure, it would be very useful to be able to stratify the risk of using different substances in each patient. Collecting such data will not be so easy, as the symptoms vary both depending on the studied population and depending on the trigger. The severity cannot be known for sure due to the interactions between generic and environmental factors. Additionally, the lethality depends on the dose, exposure time and patient’s comorbidities [88]. AI programs have a set of algorithms based on which they “make” a decision, using information from databases [86]. Because of the multiple factors and correlations with the background and genetic field of each patient, such a program would be useful to warn the doctor by displaying the anaphylaxis risk (in percentage) that the patient has, to each of the substances that they will be exposed to [88]. This kind of program will be trained effectively if it manages to continue learning while doing its job. To achieve this, after each intervention, we should tell the program to which substances the patient was exposed to, and whether or not he or she had any anaphylactic symptoms. This process is called active learning [89].

With everything noted above, one question remains: would it be possible and useful enough to develop such a software? This question is raised because anaphylactic shock is quick, and it is not accompanied by other symptoms that would indicate such an event.

Because of the large and diverse data needed, we could use the statistics of adverse drug reactions to boost the capacity of the software. Beside other uses, the AI is a very powerful data mining tool, that can be used not only for analysis and active scoring, but also for drug design, virtual screening, and quantitative structure–activity relationship analysis [90]. Going even further, in biomedical and pharmaceutical studies, AI was used to accelerate research [91]. Management of this kind of data requires versatile frameworks [89]. In drug treatment, the AI has been already studied and used to select the optimal drug or combination of drugs for individual patients, based on predictions made about drug–drug and drug–target interactions. It is true that, to be able to do that, it was necessary to be able to access a large data set that included data even about genetics or proteomics of the patient [92]. As the database would be vast, probably incomplete, and without any prodrome, it would be useful to use clustering algorithms. These algorithms are used for unsupervised classification, meaning that, the data set is broken down into clusters with data as similar as possible in an attempt to find new associations useful for machine learning [93]. After the creation of these groups, the representative group for patients with anaphylactic shock will have to be identified. More than that, with the multiple kernel density clustering, it is possible to “recover” missing data values from the dataset in attempt to obtain a database as complete as possible [94].

## 10. Conclusions

The developments in surgical and anesthetic procedures have been enormously evaluated in recent decades. Many substances are used in order to perform a safe and efficient procedure. Adverse reactions, including life-threatening reactions, may occur in a minority of patients. The surgical team must always be aware of this and a multidisciplinary team (consisting of surgeon, anesthetist, allergist) must be involved in assessing and remedying the situation. An AI program could be the key for a better perception of the risk of anaphylaxis in each patient, in order to avoid these types of events.

## Data Availability

All information presented in this review is documented by relevant references.

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
