# Peer review of "Could Artificial Intelligence Prevent Intraoperative Anaphylaxis? Reference Review and Proof of Concept"

_medicina, 2022, doi:10.3390/medicina58111530_

Round 1

Reviewer 1 Report

PUBLICATION  VERY INTERESTING COLLECTS A LOT OF ARTICLES, CONSTITUTES DE FACTO REFERENCE REVIEW AND SUMMARY OF ADVERSE REACTION RISK TO MEDICINES USED DURING ANESTHESIA OR SURGERY TREATMENTS. USUALLY A DOCTOR INTERESTED IN THE SUBJECT, SOLVING THE PROBLEM, SPENDS MANY HOURS ON ANALYSIS OF THE CAUSES OF AN ADVERSE EVENT.
 1. SUGGESTIONS FOR THE TITLE: CHANGE TO A MORE APPROPRIATE CURRENT TITLE Intraoperative events - the path to the guard of Artificial Intelligence - WITH ONE ARTICLE INTELLIGENCE CAPITAL MAY BE MISFEED, WHAT IS THE ARTICLE ABOUT?

WHAT MAY HARM THE PATIENT DURING A SURGICAL TREATMENT? - REFERENCE REVIEW

-CHALLENGES FOR DOCTORS IN THE CARE OF A PATIENT UNDER SURGERY REVIEW OF THE REFERENCES

RISK OF ADVERSE EVENTS DURING SURGERY - REVIEW OF THE REFERENCES

2. DRUG REACTIONS WE DIVIDE THE REACTIONS OF TYPE:  A, B, C, D, E, F- SKORO ALREADY RECALL THE AUTHORS ABOUT THE TYPES NEED TO REPLACE ALL 3. In addition, every adverse event related to exposure to the drug, as well -described is characterized by time and cause -and -effective dependencies, here it is worth adding demols behind Pascal, the Institute of Uppsala, causal dependence can be certain, likely, likely, possibly, doubtful, umbilical, trivial -like, etc. AND FOR THIS, AN ARTIFICIAL INTELLIGENCE WHICH WILL COLLECT THE DATA EW. EARLY REACTIONS, THE EXISTENCE OF OTHER DISEASES,

Author Response

First of all, thank you for your honest review. We have analyzed your suggestions and we announce you that we made the following modifications:

  1. We have changed the title as you suggested so that it is as appropriate as possible to the subject of the paper.
  2. We have included the ABCDEF classification in the Introduction chapter (rows 29-32).
  3. We have added more information in the AI chapter so that we can clarify the possibility of making such a software, using the clustering algorithms. These algorithms should be able to attribute how likely would be for a cluster to suffer an anaphylactic shock (rows 373-394).
  4. We have updated the bibliography with new relevant entries.

Reviewer 2 Report

The manuscript has very wide reference and is a good source of information about the possible substances, causing perioperative anaphylaxis. As expected, the reference presents very low incidence of anaphylaxis cases, which are unpredictable based only on the statistical data. The difficulties in predicting possible perioperative anaphylaxis are related to the fact, that the anaphylaxis event may occur in patients without any allergic or anaphylaxis history. It depends on many factors, pertaining mostly to the patient than the substances. In my opinion a possible AI expert system has to process data not only for the substances but also for patients and in very scrutiny way, possibly supported by allergist, which more or less eliminates the need of AI. Commenting the possible set of risk factors and ways to define them could be of real value for the manuscript. I think, that it would be good if the authors share their results of implementing any algorithms for prediction or reduction the risk of perioperative anaphylaxis, using either human or AI.

Author Response

First of all, thank you for your honest review. We understand your concerns about the possibility and utility of such a software. As a response to those concerns, we’ve made the following modifications:

  1. We have changed the title so that it is as appropriate as possible to the subject of the paper as suggested by the other reviewer;
  2. We have included the ABCDEF classification in the Introduction chapter (rows 26-33); We have added new data regarding the AI.
  3. We have added information about the clustering algorithms (rows 373-394) which might solve two of the problems raised: the vast data that might be even incomplete and the way how the classification can be worked out, in a non-supervised way (because as you said, anaphylactic shock can happen even to people that never expressed any allergies).
  4. We have updated the bibliography with new relevant entries.
